# Regional clozapine, ECT and lithium usage inversely associated with excess suicide rates in male adolescents

Adrian E. Desai Boström [1,2,3,4] ✉, Peter Andersson [5,6], Mathias Rask-Andersen [7], Håkan Jarbin [8,9], Johan Lundberg [3,4] & Jussi Jokinen [1,4]

Advanced psychiatric treatments remain uncertain in preventing suicide among adolescents. Across the 21 Swedish regions, using nationwide registers between 2016–2020, we found negative correlation between adolescent excess suicide mortality (AESM) and regional frequencies of clozapine, ECT, and lithium (CEL) usage among adolescents ($\beta = -0.613$, $p = 0.0003$, 95% CI: $-0.338$, $-0.889$) and males ($\beta = -0.404$, $p = 0.009$, 95% CI: $-0.130$, $-0.678$). No correlation was found among females ($p = 0.197$). Highest CEL usage among male adolescents was seen in regions with lowest quartile (Q1) AESM ($W = 74$, $p = 0.012$). Regional CEL treatment frequency in 15–19-year-olds was related to lower AESM in males, reflecting potential treatment efficacy, treatment compliance or better-quality mental health care. Suicide prevention may benefit from early recognition and CEL treatment for severe mental illness in male adolescents. The results indicate association but further research, using independent samples and both prospective and observational methodologies, is needed to confirm causality.

Worldwide, suicide is a major cause of death, years-lost and an important public health concern. Globally, it is estimated that 800,000 deaths each year are caused by suicide[1]. Despite declining rates observed over recent decades[2], suicide is yet the leading cause of death worldwide among 15–24-year-olds[3]. There is regional variation. For example, the rate of confirmed deaths by suicide in 15–19-year-olds is higher in Sweden than in comparable countries (6.9 per 100,000 inhabitants in this age-group compared to 3.7 and 4.7 for Denmark and Germany, respectively). Importantly, only 36% of Swedish adolescent suicide victims (48% and 27% for females and males, respectively) were registered in psychiatric care facilities the year prior to their death[4].

Across the total lifespan, death by suicide is more common in males compared to females. Data from the Global Disease Burden Study indicates that this pattern is established in young adulthood. The estimated global annual mortality rates from suicide in the 15–19 age-range amount to 8.5 and 8.2 per 100,000 in females and males, respectively. In young adulthood (20–24 age-range), reported estimates diverge by sex – i.e., 10.2 and 16.2 per 10,000 females and males, respectively[2].

Contemporary recommendations to reduce adolescent suicide rates emphasize psychotherapeutic treatment (e.g. dialectical behavior therapy and mentalization-based treatment), and community and

[1]Department of Clinical Sciences/Psychiatry, Umeå University, Umeå, Sweden. [2]Department of Women's and Children's Health/Neuropediatrics, Karolinska Institutet, Stockholm, Sweden. [3]Stockholm Health Care Services, Stockholm, Sweden. [4]Centre for Psychiatry Research, Department of Clinical Neuroscience, Karolinska Institutet & Stockholm Health Care Services, Region Stockholm, Karolinska University Hospital, SE-171 76 Stockholm, Sweden. [5]Division of Psychology, Department of Clinical Neuroscience, Karolinska Institutet, Stockholm, Sweden. [6]Centre for Clinical Research, Dalarna, Uppsala University, Falun, Sweden. [7]Department of Immunology, Genetics and Pathology, Uppsala University, Uppsala, Sweden. [8]Department of Clinical Sciences Lund, Section of Child and Adolescent Psychiatry, Lund University, Lund, Sweden. [9]Child and Adolescent Psychiatry, Region Halland, Halland, Sweden. ✉e-mail: adrian.desai.bostrom@ki.se

school based early prevention initiatives[3,5–7]. The potential role of psychopharmacologic interventions for suicide prevention in adolescent populations appears more conflicting. For example, early epidemiological reports[6] pointed to an inverse association between prescription rates of selective serotonin reuptake inhibitors (SSRI:s) and youth suicide rates[8]. Later reviews, however, emphasized a lack of evidence for the efficacy of psychopharmacologic interventions in adolescent populations, and drew on meta-analytic data from randomized-controlled trials to implicate putative links between antidepressant drug usage and increased suicidality in youths[5,9] – concerns that Lagerberg et al. recently impugned as resulting from selection bias confounding[10]. Notably, several recent reports found contrasting findings[11].

Severe mental illness is the most influential and preventable predictor of death by suicide[3]. Reviews of descriptive epidemiological reports estimate that psychiatric disorders are prevalent in up to 90% of unselected adolescent suicide victims, where the presence of any affective disorder (including major depressive disorder and bipolar disorder) is prevalent in 44–76% of suicide cases[12]. Electroconvulsive therapy (ECT), lithium and clozapine are effective treatments for the most severe forms of major depressive disorder, bipolar disorder, and schizophrenia. Albeit inconclusive[13], these therapies have been previously associated with reductions in suicide mortality in adults[14–16]. Although it appears that previous work has not looked at these interventions with regard to suicide mortality in adolescents[3], there have been studies demonstrating effectiveness of these treatment regimens in reducing suicidal behavior in adolescents[17–19]. The Course and Outcome of Bipolar Youth study reported on the longitudinal course in a group of 413 children and adolescents (age span 7–17.11 years old) with bipolar disorder during a mean follow-up of 10 years. In this sample, a halving of non-fatal suicide attempts (80 such events reported in the total sample) was observed in the lithium treated group compared to patients treated with other mood stabilizer drugs[17]. Similarly, a retrospective chart review of 54 adolescents (mean age 15.8 year old) treated with ECT for refractory mood disorders - reporting on a 52.8% response rate (defined as a Clinical Global Impressions [CGI] score ≤2) – observed reductions in suicidal ideation and self-injurious behavior following the index course of ECT (mean number of treatments=13.7 ± 6.3)[18]. Moreover, a Danish population-based study conducted between 1994 to 2006 and reporting on 662 cases of early onset schizophrenia (EOS) observed that a history of previous suicide attempt predicted initiation of clozapine treatment in the real-world management of pediatric populations with EOS. 96 out of 108 clozapine treated youth (88.9%) redeemed prescriptions for at least six consecutive months, indicating a high rate of tolerability in real-world clinical cases. Clozapine discontinuers ($n = 12$) had a higher rate of nonfatal suicide attempts and there were preliminary indications that participants continuing treatment past six months exhibited a more favorable clinical response with shorter hospital stays and better occupational outcomes at age 20 compared to discontinuers[19]. Whilst one might argue that the prevention of suicidal behavior, more favorable outcomes and shorter hospital stays has implications for adolescent suicide mortality, the high ratio of suicidal behavior (i.e., self-harm, suicidal ideation and nonfatal suicide attempts) to suicide deaths – estimated to be 50:1 to 100:1[20–22] – warrant further investigation in relation to suicide mortality.

Using real-world Swedish registry data across the 21 Swedish regions (2016–2020) and implementing rigorous procedures to reduce putative confounding – in a sample encompassing 632 confirmed suicide deaths (200 of which in 15–19-year-olds) – the present study investigated associations between regional adolescent excess suicide mortality (AESM) and treatment usage frequencies of clozapine, ECT and lithium (CEL) in 15–19-year-olds ($n = 21$). The time-period was chosen to match the major course direction in national treatment guidelines regarding ECT – since 2016 and with highest priority

recommending its use in the care of post-pubertal adolescents with severe MDD with mood-congruent psychotic symptoms, catatonia, or treatment resistance. Similarly, information on regional suicide death rates was available up to 2020, hence, 2021–2022 treatment frequencies were not included in the analysis. The relative contribution of each treatment modality to the proxy variable measuring mean treatment usage frequencies was estimated. To reduce putative influence of regional differences in, for example, population size, socioeconomic status, substance abuse rates, or availability and quality of psychiatric care, the primary outcome variable AESM was defined as the regional mean across 2016–2020 of normalized year-wise differences in the number of suicide deaths per 100 000 inhabitants in adolescence/pubescence (15–19-year-olds) compared to suicide deaths in young adulthood (20–24-year-olds).

## Results

### Baseline characteristics of data

Each treatment modality contributed equally to the derived proxy variable in the combined sexes group (fraction = 1.0). Clozapine and ECT variables exhibited predominant influence on the proxy variable in the female and male subgroup, respectively (fractions: 1.31 and 1.86, respectively). There were large variations in the combined sexes group across regions in min-max normalized mean adolescent usage frequencies across 2016–2020 for ECT (ages 13–17), lithium and clozapine (both ages 15–19), i.e., ECT (IQR: 0.00 to 0.34), lithium (IQR: 0.39 to 0.53) and clozapine (IQR: 0.20 to 0.46). Likewise, there were substantial variations in the mean treatment usage frequencies across 2016-2020 (encompassing values for all studied treatment modalities); i.e., IQR: 0.27 to 0.38. Disparities are perhaps best illustrated by values at the extreme, i.e., for ECT (min: 0.00, max: 0.55), lithium (min: 0.28, max: 0.73), clozapine (min: 0.00, max: 0.60) and their mean (min: 0.09, max: 0.56). Thus, variations between regions are larger in the case of clozapine compared to lithium. The national non-normalized mean adolescent suicide death rates per 100,000 inhabitants across 2016–2020 was near 2-fold higher in male (9.14) participants when compared to females (5.18). By the same analysis, mean suicide death rates across 2016–2020 were more than doubled in young adulthood – albeit exhibiting the same proportion between sexes (i.e., in males [18.66] and females [9.24], respectively). Moreover, there were large variations between regions in min-max normalized baseline adolescent suicide rates for females (IQR: 0.11 to 0.68) and males (IQR: 0.29 to 1.26), respectively. By the same analysis, suicide rates in young adulthood were comparable for females but not for males, i.e., females (IQR: 0.20 to 0.89) and males (IQR: 0.78 to 1.80). Similarly, sex-dependent variations between counties in excess adolescent suicide death rates were largely comparable, i.e., for females (IQR: 0.47 to 0.56) and males (IQR: 0.45 to 0.57), respectively. Derived proxy-variables for advanced treatment usage and excess adolescent suicide deaths were compared across the sex groups by independent samples $t$-tests, evincing no specific sex-dependent associations to these variables ($p$-value = 0.808 and $p$-value = 0.536, respectively).

### Associations between Mean Treatment Usage and Excess Adolescent Suicide Death Rates across 2016–2020

On the association analysis performed in the combined sexes group – using robust linear regression models with population-weights, MM-estimates, and otherwise recommended settings – mean clozapine, ECT and lithium treatment usage frequencies was inversely associated with excess adolescent suicide death rates across Swedish regions for the period 2016–2020 (β = −0.613, $p$-value = 0.0003, 95% CI: −0.338, −0.889)(Fig. 1). The data was subsequently stratified by sex and analyzed separately for females and males, respectively. By the same analyses, no confirmed association was observed for the female subgroup (β = −0.149, $p$-value = 0.197, 95% CI: −0.366, 0.069). For males, however, derived treatment usage frequencies were inversely

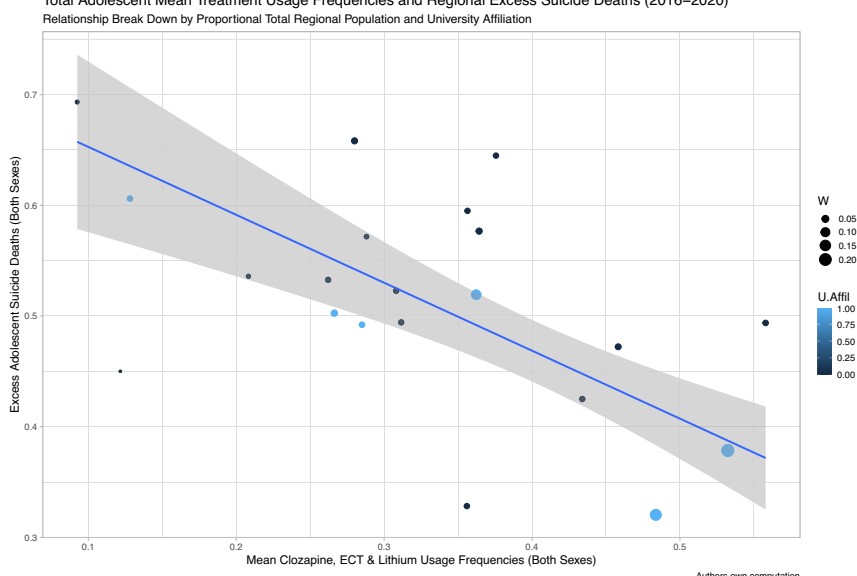

**Fig. 1 | Mean Treatment Usage Frequencies and Regional Excess Suicide Deaths across 2016–2020 [Both sexes].** The Y-axis depicts the regional mean of min-max normalized year-wise differences in suicide deaths per 100,000 inhabitants between adolescence and young adulthood. The X-axis depicts regional mean values of min-max normalized clozapine, ECT and lithium treatment usage frequencies across 2016–2020 values. The slope and confidence intervals (CI:s) of the robust linear regression model contrasting these two variables are depicted as a blue line (slope) with grey shading (CI:s). Region population in relation to the national population is illustrated by the circle diameter, and counties affiliated with medical universities are highlighted in blue. These include Skåne region with Lund University, Stockholm region with Karolinska Institutet, Uppsala region with Uppsala University, Västerbotten region with Umeå University, Västra Götaland region with Gothenburg University/Sahlgrenska Academy, and Östergötland region with Linköping University. The figure demonstrates that regional mean of min-max normalized clozapine, ECT and lithium usage frequencies across 2016–2020 are inversely correlated with regional excess adolescent suicide deaths in the combined sexes group ($\beta = -0.613$, p-value = 0.0003, multiple R-squared: 0.123, adjusted R-squared: 0.077, 95% CI: −0.338, −0.889). Abbreviations: 95% CI, 95% confidence interval; ECT, electroconvulsive therapy; U.Affil, medical university affiliation (regions affiliated to medical universities are coloured in blue and regions unaffiliated to medical universities are coloured in black); W, weights (regional population size expressed as a percentage of the total national population).

associated to excess adolescent suicide deaths ($\beta$ = -0.404, p-value=0.009, 95% CI: −0.130, −0.678)(Fig. 2). In a validation analysis, we aimed to confirm that counties with lower excess adolescent suicide mortality exhibited higher mean treatment usage frequencies. Thus, lower-quartile (Q1) counties based on excess adolescent suicide rates were dichotomized (lower quartile meaning counties with less excess adolescent suicide rates compared to Q2-Q4) and contrasted to regional mean treatment usage frequencies, separately for the combined sexes group and males, respectively. The one-sided Wilcoxon rank sum exact test confirmed these associations in the combined (W = 67, p-value = 0.047) and male (W = 74, p-value = 0.012) group, respectively (Fig. 3). As no significant associations were revealed in the main analysis for the female subgroup, this group was not subjected to such post-hoc analysis.

**Post-hoc analyses investigating associations between single treatment usage and excess adolescent suicide death rates across 2016–2020**

As a post-hoc analysis to determine associations of any single treatment modality to excess adolescent suicide death rates, we first compared individual treatment variables that were not strongly correlated amongst themselves ($r < 0.5$) with the excess adolescent suicide death rate variable – separately in each sex group. Strongly correlated treatment variables were averaged, and this mean value was implemented as co-variate in the subsequent analyses. In the combined sexes group, mean ECT and lithium usage frequencies were inversely associated to excess suicide rates in adolescents ($\beta$ = −0.14, p-value = 0.004) in the multivariate robust linear regression model. The one-sided Wilcoxon rank sum exact test confirmed this association in the case of lithium (W = 71, p-value = 0.022, Bonferroni-adjusted

p-value = 0.045), whereas ECT exhibited a non-significant trend in the same direction (W = 65.5, p-value = 0.05, Bonferroni-adjusted p-value = 0.101) – i.e., indicating that regions with lower adolescent suicide death rates exhibited higher lithium and ECT usage frequencies (Supplementary Fig. 2). In the subgroup-analyses, female ECT usage frequencies were inversely associated to excess female adolescent suicide rates ($\beta$ = −0.04, p-value = 0.005, Bonferroni-adjusted p-value = 0.016, 95% CI: −0.062, −0.014) in the univariate robust linear regression models (Supplementary Fig. 3) – an association that was confirmed by the one-sided Wilcoxon rank sum exact test (W = 67, p-value = 0.039). Similarly, male ECT usage frequencies were negatively associated with excess male suicide death rates ($\beta$ = −0.06, p-value = 0.016, 95% CI: −0.1, −0.015) in the multivariate robust linear regression model performed in females – findings that were validated in subsequent nonparametric tests (W = 68, p-value = 0.022)(Supplementary Fig. 4).

## Discussion

This study demonstrates that regional clozapine, ECT and lithium usage frequencies in 15–19-year-olds are associated with reductions in excess regional suicide death rates in male adolescents. Specifically, post-hoc analyses of individual treatments demonstrated associations between regional pubescent lithium usage and lower AESM – whereas adolescent ECT usage rates exhibited a non-significant trend in the same direction. It could be argued that the observation of an inverse association between regional CEL-treatment utilization rates and excess suicide death rates in male adolescents adds to the relevance of these findings, as adolescents receiving CEL-treatment – typically reserved for especially severe or treatment-refractory cases of severe mental illness - would be expected to elicit increased suicide deaths.

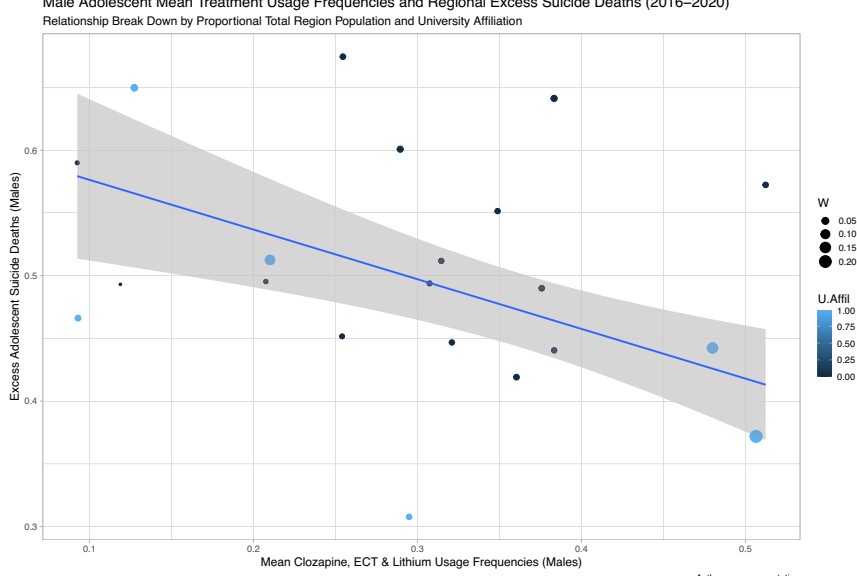

**Fig. 2 | Mean Treatment Usage Frequencies and Regional Excess Suicide Deaths across 2016-2020 [Males].** The Y-axis depicts the regional mean of min-max normalized year-wise differences in suicide deaths per 100,000 inhabitants between adolescence and young adulthood. The X-axis depicts regional mean values of min-max normalized clozapine, ECT and lithium treatment usage frequencies across 2016–2020 values. The slope and confidence intervals (CI:s) of the robust linear regression model contrasting these two variables are depicted as a blue line (slope) with grey shading (CI:s). Region population in relation to the national population is illustrated by the circle diameter, and counties affiliated with medical universities are highlighted in blue. These include Skåne region with Lund University, Stockholm region with Karolinska Institutet, Uppsala region with Uppsala University, Västerbotten region with Umeå University, Västra Götaland region with Gothenburg University/Sahlgrenska Academy, and Östergötland region with Linköping University. The figures demonstrate that regional mean of min-max normalized clozapine, ECT and lithium usage frequencies across 2016–2020 are inversely correlated with regional excess adolescent suicide deaths in males (β = −0.404, *p*-value=0.009, multiple R-squared: 0.018, adjusted R-squared: −0.034, 95% CI: −0.130, −0.678). Abbreviations: 95% CI, 95% confidence interval; ECT, electroconvulsive therapy; U.Affil, medical university affiliation (regions affiliated to medical universities are coloured in blue and regions unaffiliated to medical universities are coloured in black); W, weights (regional population size expressed as a percentage of the total national population).

Our findings are based on openly available data from the Swedish National Board of Health and Welfare and the Swedish ECT registry (coverage rate >90%) across 2016–2020, encompassing all registered Swedish citizens that in the years 2016–2020 were aged 15–19 or 20–24-years-old and included 632 confirmed suicide deaths (200 of which pertaining to adolescents). The data was processed and normalized over the years 2016–2020 and subsequently analyzed and compared between the 21 Swedish regions (*n* = 21). The aggregated nature of these observational data precludes a causal interpretation of the results. Regional CEL-treatment frequencies are associated with reduced AESM in 15–19-year-old males and may reflect effects relating to treatment efficacy, treatment compliance or better-quality mental health care at the regional level. Nevertheless, these results add to the literature indicating potential gains from early recognition and CEL-treatment of severe mental illness for suicide prevention in adolescent populations and warrant the conduct of high-quality replication studies in independent samples, utilizing both prospective methodologies and large sample observational research.

The results of this study provide additional support for the previous inconclusive evidence on the suicide-protective effects of CEL-treatments in adult populations[13] and extend this research to examine the impact of these interventions on suicide mortality in adolescents. Despite prior studies demonstrating reductions in suicidal behavior and nonfatal suicide attempts from CEL-treatments in adolescents[17–19], the relationship between these proxy outcomes and actual suicide deaths is limited, with estimated concordance rates of only 1-2%[20–22]. This highlights the need for further investigation into the potential relationship between CEL-treatments and suicide mortality. The present study provides important knowledge on this topic and suggests a plausible connection, albeit further research is necessary before causal inferences can be confidently made.

Previous studies described substantial sex differences in suicide-related outcomes in both adolescent[12] and adult populations, and implicated distinct underlying neurobiological mechanisms[23]. For example, suicide death rates are generally higher for males compared to females in all countries allowing for the systematic collection of such data (with the exception of China)[12] – findings that were confirmed in the present study. These differences between the sexes in suicide death rates are believed to be conferred by the higher prevalence of multiple suicidal risk factors in male adolescents, i.e., implementing more lethal suicide attempt methods, exhibiting higher levels of comorbid mood and substance abuse disorders and more aggression[12]. The inability of the present study to discern significant suicide-protective associations in stratified analyses of females and clozapine could be reflective of lower power in these subpopulations (males were generally overrepresented in baseline suicide rates in both adolescence and young adulthood by factor ~2 and adolescent treatment-refractory schizophrenia spectrum disorder - when compared to major affective disorders - would be expected to contribute less to suicide death rates[12]). Other potential explanatory factors pertain to study design limitations (i.e., assessing excess suicide rates as opposed to baseline values), biological differences, or other factors. Our findings rely on the assumption of comparable within-regional psychiatric care affiliation rates of suicide victims in adolescence and young adulthood. Confirmatory analyses taking such factors into account – as well as replication of these findings across independent data sets – would be of value. Data availability issues precluded the possibility to explore the potential moderating role on the observed associations of clinical characteristics of individual adolescent recipients of the studied treatment modalities, dosages of lithium or clozapine and the number of ECT treatment courses administered to individual patients. Moreover, while the study did not account for

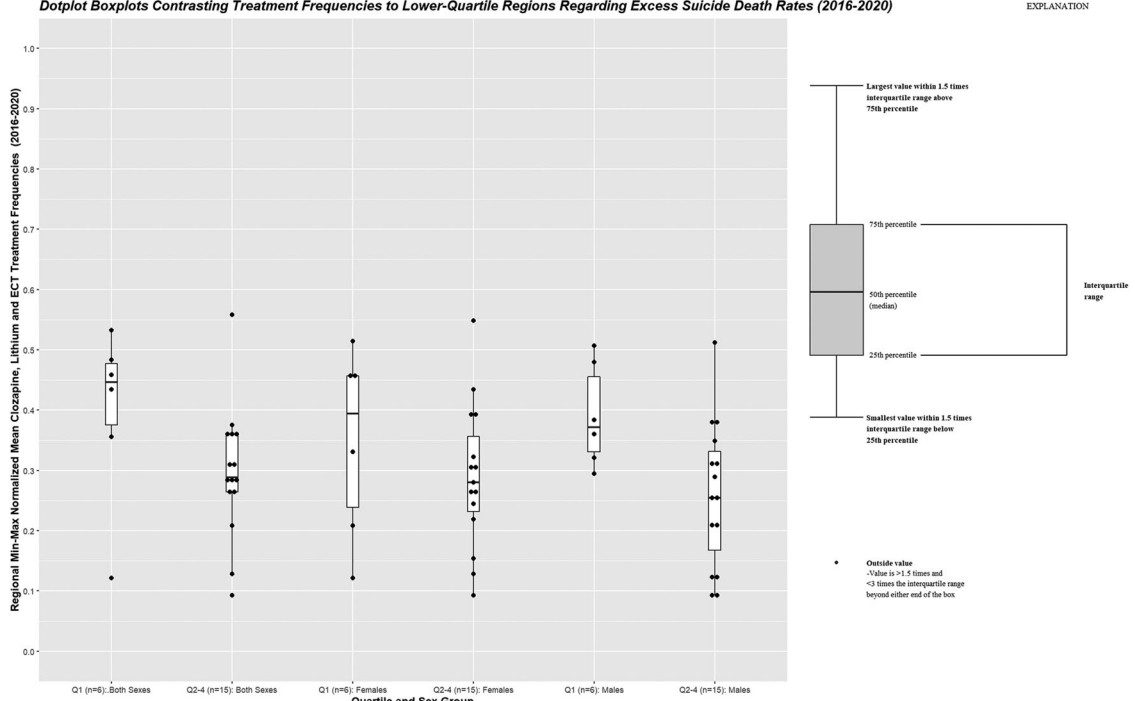

**Fig. 3 | Mean Treatment Usage Frequencies and Regional Excess Suicide Deaths across 2016–2020.** The Y-axis depicts regional mean values of min-max normalized clozapine, ECT and lithium treatment usage frequencies across 2016–2020 values (derived in Section 2.1 (1)). The X-axis depicts the lower (Q1, n = 6) and other (Q2-Q4, n = 15) counties regarding mean of min-max normalized year-wise differences in suicide deaths per 100,000 inhabitants between adolescence and young adulthood (variables derived in Section 2.1 (1), separated by sex group (both sexes, females, and males, respectively). Thus, per definition, Q1-counties exhibited lower excess adolescent suicide deaths in comparison to Q2-Q4. Mean treatment usage frequencies were compared between the in-silico generated subgroups in using the one-sided Wilcoxon rank sum exact test – performed separately for each sex group. Lower-quartile counties regarding excess adolescent suicide deaths were

associated with higher mean treatment usage frequencies in the combined sexes (W = 67, p-value=0.047; Q1 - minima: 0.122, maxima: 0.532, centre [median]: 0.446, bounds of box: 0.375–0.477, lower whisker: 0.356, upper whisker: 0.532; Q2-Q4 - minima: 0.093, maxima: 0.558, centre [median]: 0.288, bounds of box: 0.264–0.359, lower whisker: 0.128, upper whisker: 0.376) and male (W = 74, p-value = 0.012; Q1 - minima: 0.122, maxima: 0.515, centre [median]: 0.394, bounds of box: 0.239–0.457, lower whisker: 0.122, upper whisker: 0.515; Q2-Q4 - minima: 0.093, maxima: 0.549, centre [median]: 0.28, bounds of box: 0.232–0.357, lower whisker: 0.093, upper whisker: 0.434) groups, respectively. The female group evinced no such associations.ECT electroconvulsive therapy, Q1 first quartile; Q2-Q4 second, third and fourth quartile.

comorbid illness and access to other therapies, this would not necessarily reduce confidence in the observed associations – treatment with clozapine and ECT are usually reserved for especially severe or treatment-refractory cases of schizophrenia or MDD (unresponsive to auxiliary treatments) where the co-occurrence of any comorbid conditions precipitating suicide deaths would be expected to be higher (the opposite association was observed, i.e., regions with high CEL treatment frequencies exhibited lower AESM). In addition, the association between ECT treatment and ASEM – markedly stronger in sex-specific sub analyses compared to the combined sex group – is explained by downstream effects from within-regional variability in year-wise ECT treatment utilization frequencies (the data normalization method was chosen for its ability to incorporate both year-wise regional variability and baseline magnitude). The lack of significance in the combined sex group (albeit exhibiting a non-significant trend in the same direction) reduce confidence in this association. Furthermore, downstream effects from the 2020 onset of the Covid-19 pandemic[24] has been hypothesized to increase mental health problems in youth, potentially influencing suicide death rates. The present study evinced no such effects in males or females for the year 2020 compared to 2016–2019. Lastly, it must be acknowledged that suicide in adolescence is a complex topic where much remains to be understood, arguably with many potential pathways leading to such tragic outcomes. For example, there are some factors that might especially influence suicide risk in adolescents, such as, for example, emotional dysregulation, substance use and misuse[25]. Thus, one size does not seem to fit all, and an emphasis on multimodal and multilevel

preventative efforts seems warranted. Importantly, addressing low psychiatric care affiliation rates of Swedish adolescent suicide victims[4] would be a pre-requisite to any successful suicide preventative effort.

In conclusion, using rigorous statistical methods to analyze data across Swedish regions in 2016–2020, we demonstrate that regional clozapine, ECT and lithium usage frequencies in 15–19-year-olds are inversely associated with excess suicide deaths in male adolescents. Importantly, our data demonstrated substantial regional variations in population-adjusted estimates of overall clozapine, ECT and lithium treatment usage frequencies in adolescents. Thus, current frameworks for suicide prevention in adolescent populations, emphasizing psychotherapy and school and community-based efforts[3,13], may benefit from also considering CEL treatments for severe mental illness in male adolescents. These findings rely on the assumption of comparable psychiatric care affiliation rates of suicide victims in adolescence and young adulthood within regions. To emphasize the importance of addressing the issue of suicide in adolescent populations, it is worth noting that confidential support is available to those in crisis or emotional distress through the website https://findahelpline.com/i/iasp.

## Methods
### Data sources and initial processing
Openly available data from the Swedish National Board of Health and Welfare (available in Swedish: https://sdb.socialstyrelsen.se/if_dor/val. aspx; https://sdb.socialstyrelsen.se/if_lak/val.aspx)[26,27] was extracted

for 21 Swedish counties across 2016–2020 in the age-ranges 15–19 and 20–24, respectively, and for the following variables: The number of dispensations to adolescents recorded for lithium (ATC-code N05AN01) and clozapine (ATC-code N05AH02) per 1000 inhabitants (based on population estimates from January 1st of the recorded year), and confirmed suicide death rates per 100,000 inhabitants. All data was extracted for both sexes, males, and females, respectively – across all Swedish counties for the years 2016–2020. Similarly, openly available data from the Swedish ECT registry (available in Swedish: https://ect.registercentrum.se/statistik/utdata-ect/p/rJ3vhF3Lw)[28] detailing the number of patients receiving ECT treatment (age-group and sex) was extracted for the years 2016–2020 and each region in the age-range 0–17 (<18). All individuals receiving ECT treatment were aged 13-years-old or older and the overwhelming majority of the sample consisted of individuals aged 15–17-years. The rationale for including the years 2016–2020 and the included variables are explained in detail in Supplementary Material, where sourced data, analysis code and other important information has also been made available to facilitate the expedient replication of our results.

## Statistical considerations

Several measures were taken with the aim of strengthening robustness of included variables. These initial steps were performed in using Microsoft Excel for Microsoft 365 MSO (Version 2204 Build 16.0.15128.20278) and are explained in greater detail in Supplementary Materials.

## Min-max normalization to account for both baseline magnitude and variability in regional year-wise data

In brief, mean and median values for each region and sex-group across 2016–2020 exhibited greater standard deviation than mean/median value for a majority of key variables and, thus, considered unrepresentative of the underlying data distribution. Therefore, min-max normalization was implemented across each year, region, and treatment type (i.e., for example, $\text{DerivedValue}_{2016} = \text{Value}_{2016}\text{-min}(\text{Value}_{2016-2020})/(\max(\text{Value}_{2016-2020})\text{-min}(\text{Value}_{2016-2020}))$. In the case of suicide death rates, the difference in suicide rates in adolescence and young adulthood per 100,000 inhabitants was calculated for each region, year, and sex group (i.e., $\text{DerivedExcessSuicideRate Adolescence}_{2016} = \text{SuicideRateAdolescence}_{2016}-\text{SuicideRate YoungAdulthood}_{2016}$) and subsequently subjected to min-max normalization. The mean of min-max normalized values across 2016–2020 was implemented in the subsequent analysis, whereby each region and sex-group were represented by a mean treatment value measured across 2016–2020 and one value for excess adolescent suicide deaths. When compared to the median of min-max normalized values, the mean was considered more representative of the underlying data distribution by, for example, also recognizing counties providing treatment in only one or two years in the studied five-year-period. The adequacy of the min-max normalized values to not only upregulate regions exhibiting low variability between years, but also to account for the baseline magnitude of treatment usage frequencies – the derived treatment proxy variable was investigated by Pearson correlations to the mean year-wise sum of baseline values for clozapine, ECT and lithium – exhibiting a medium to strong positive correlation ($r = 0.49$).

## Subtraction of suicide death rates within regions and between age-groups to reduce putative confounding

To reduce effects of potential unmeasured sources of confound on suicide rates across counties, the primary outcome variables were based on subtraction between suicide death rates in adolescence and young adulthood. Thus, the influence of regional differences on suicide death rates in, for example, population size, socio-economic status, substance abuse, or availability and quality of

psychiatric care, could thus largely be reduced. Similarly, subsequent min-max normalization contributed to normalizing potential confounding effects of substantial outliers regarding baseline absolute values.

## Input variable considerations

Basing lithium and clozapine values on the number of dispensations reduced influence from short-term treatment and min-max normalization contributed to downregulating potential confound from regions exhibiting one or two years with extremely high values and zero (no treatment) for the other years, neither considered conciliable with best-practice care.

## Rationale for the selected study time-period

The time-period was chosen to match the major course direction in national treatment guidelines regarding ECT – since 2016 and with highest priority recommending its use in the care of post-pubertal adolescents with severe MDD with mood-congruent psychotic symptoms, catatonia, or treatment resistance. Similarly, at the time of data procurement, information on regional suicide death rates was available up to 2020, hence, 2021–2022 treatment frequencies were not included in the analysis.

## Integrating clozapine, ECT and lithium usage frequencies

Implementing mean treatment values of clozapine, ECT and lithium usage frequencies should further increase robustness, while also allowing for the recognition of ECT-aversive (or clozapine or lithium-aversive) counties that may provide adolescents with treatment with other studied modalities when indicated.

## Implementation of weighted robust regression models to further reduce confounding

The usage of robust linear regression models weighted to regional population estimates, contributed to reducing confound from outlier counties with small populations (whereby the influence of a single clinician could be expected to significantly alter measured frequencies – while possibly unrepresentative of the overall quality of care provided in the region).

## Statistical analysis

Initial data processing (i.e., calculation of min-max normalized values) was performed using Microsoft Excel365 MSO [Version 2210 Build 16.0.15726.20188] 64-bit). All downstream statistical analyzes were performed using R version 4.0.3. and are explained in greater detail in Supplementary Material. To sum up, all variables were investigated for normal distribution. Variables not satisfying requirements for normal distribution were subjected to transformation by Blom's method[29] for the subsequent robust linear regression analyses (but not for non-parametric tests, i.e., the Wilcoxon rank sum exact test). To examine potential confound conferred from the 2020 onset of the Covid-19 pandemic[24], we implemented the Wilcoxon Rank Sum Exact test to compare the mean of min-max normalized values for excess adolescents suicide death rates over 2016–2019 with values in the year 2020 across the 21 regions – evincing no significant difference in any sex group (combined sexes: $p$-value = 0.811, females: $p$-value = 0.613, males: $p$-value = 0.358). Multiple ordinary least squares regression was implemented to examine the relative contribution of each treatment modality to the mean clozapine/ECT/lithium proxy variable. Main analyses of associations between regional excess adolescent suicide mortality and mean clozapine/ECT/lithium treatment usage frequencies were investigated by robust linear regression models using the R-package 'robustbase'[30], specifying recommended setting (KS2014), standard MM-regression estimators (guaranteeing an acceptable compromise between high breakdown (i.e., 50%) and very high efficiency (i.e., 95%)[31]) and adjusting for weights (2020 regional

population in proportion to national estimates the same year)[32]. Models exhibiting *p*-values for the primary explanatory variable <0.05 were considered significant. Main models were illustrated by x–y scatterplots, with the estimated slope coefficient from the regression model (Fig. 1). To validate significant models, the hypothesis was tested whether counties with lower quartile (Q1) excess adolescent suicide mortality exhibited greater advanced treatment usage frequencies in adolescents compared to Q2-Q4 counties, using the one-sided Wilcoxon rank sum exact test. *P*-values < 0.05 were considered significant. Post-hoc analyses were subsequently performed to determine associations in sex and treatment-stratified groups (i.e., males and females, and clozapine, ECT and lithium). Treatment variables were investigated for collinearity, and any such associations were adjusted for in downstream analyses. To reduce potential bias from overfitting of the model (given the small sample size, *n* = 21 regions), treatment variables were tested separately for the female subgroup – and resulting significance values were subjected to stringent Bonferroni-correction[33]. As in the main analysis, significant associations were validated by contrasting the dichotomized 25th quartile based on excess suicide death rates to the candidate treatment variable. In these post-hoc analyses of non-Blom-transformed values, *p*-values <0.05 (Bonferroni-adjusted in the case of females) were considered significant.

### Reporting summary
Further information on research design is available in the Nature Portfolio Reporting Summary linked to this article.

## Data availability
Raw and processed data generated in this study have been deposited in its entirety in the Open Science Framework (OSF) repository, openly accessible through the following web-links: https://osf.io/ygevm (raw and processed data, written for Microsoft Excel365 MSO [Version 2210 Build 16.0.15726.20188] 64-bit). This data was retrieved from openly accessible data repositories provided by the Swedish National Board of Health and Welfare and the Swedish ECT registry. Web links from which the data can be collected from its original sources for replication or other research purposes are provided in Methods section - 'Data Sources and Initial Processing'.

## Code availability
The analysis code is provided in Supplementary Materials and has been deposited in its entirety in the Open Science Framework (OSF) repository, openly accessible through this web-link: https://osf.io/64er9 (R-code, written for R version 4.2.0).

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

## Acknowledgements

We are grateful to the Swedish National Board of Health and Welfare and the Swedish National Quality Registry for ECT. Jussi Jokinen received funding from the Swedish Research Council (grant no. 2020-01183).

## Author contributions

A.E.D.B. conceptualized the study. A.E.D.B., H.J. and J.L. curated the data. A.E.D.B. performed all statistical analyses. M.R-A. verified the data and statistical analyses. A.E.D.B. and P.A. wrote the first draft of the manuscript. All authors contributed to the interpretation of results. All authors contributed to investigation, interpretation, review and editing of the manuscript. All authors read and approved the final version of the manuscript.

## Funding

## Competing interests

The authors declare no competing interests.
