## [Peer Review File · Nature Communications]

Reviewers' Comments:

Reviewer #1:

Remarks to the Author:

Using real-world Swedish registry data across the 21 Swedish regions (2016-2020) and implementing rigorous procedures to reduce putative confounding – in a sample encompassing 632 confirmed suicide deaths (200 of which in 15-19-year-olds) –, in the present study the Author investigated associations between regional adolescent excess suicide mortality (AESM) and treatment usage frequencies of clozapine, ECT and lithium (CEL) in 15-19-year-olds. Overall, I found this study timely, original, well conducted and scientifically sound. The Authors incorporated all my suggestions in the revised paper. I have no further comments or concerns on it. I recommend acceptance.

Reviewer #2:

Remarks to the Author:

I appreciate the authors' responses to reviewers' comments. Overall, I think the authors sufficiently addressed my concerns. This manuscript presents important information with clinical and potential policy relevance.

The authors provide a detailed reply to my comment, "I also encourage the authors include more cautions regarding the correlative nature of their study. The suicide prevention potential of these therapeutics cannot be arbitrated by this correlative study." In the authors' reply, they note that "we are concerned that the reviewer's comments may be overstated." I respectfully disagree. My comment was not that observational data do not have value. Instead, my comment was meant to highlight that the authors -- in my opinion -- needed to do a better job of contextualizing for the reader the methodology of their study and the claims made thereon. This is especially important for a prominent outlet that garners substantial press coverage. I maintain that a single correlative study cannot arbitrate the suicide prevention potential of these therapeutics. No single study is dispositive, regardless of methodology. I offer this additional context so that the authors and editors know that I do think their findings have value. In their revision, I believe the authors adequately contextualized their findings. Thank you to the authors for this undertaking.

Reviewer #3:

Remarks to the Author:

The authors have undertaken an analysis of $n=21$ regions in Sweden to identify associations between a collection of treatments (clozapine, ECT, and lithium usage rates) and suicide mortality in those regions. The authors have clearly noted the limitations of the study since they are unable to identify what overlap there is between the treatments and the outcomes because both rates arise from aggregated data. A creative approach to partially account for potential confounding is undertaken by subtracting the rates of suicides in older adolescents (15-19 year olds) from those in young adulthood (20-24 year olds) [page 7, lines 165-169].

I have had the opportunity to review the manuscript and supplementary materials, including the code and data. Overall these materials, methods, data, and code were clear and comprehensive. One comment: the way that the data were made accessible in R involved cutting and pasting values which wasn't an ideal workflow.

I believe that the paper is a valuable contribution to the literature despite the serious limitations due to its design. My main comments relate to ways to improve the presentation and reporting.

1. Title: I would encourage to authors to consider adding "Regional" before "Clozapine, ECT, and Lithium Usage Rates" in the title.
2. Reporting of p-values: I would encourage the authors to report p-values with two or three digits unless they are very small (e.g., $p < 0.0001$) rather than use $p < 0.5$ or $p > 0.10$ or $p = 0.04742$. Same for the Beta on page 11 (Beta = -0.4040600).
3. Reporting of sample size: it's important for the authors to clearly remind the reader that this analysis is effectively $n = 21$ (despite the large populations involved).
4. Discussion: add "regional" between "excess" and "suicide death rates", page 13.
5. Figure 2 would be improved if the axis labels included the sample size (e.g, Q1: Both sexes, $n = XX$). The violin plots are misleading here: wouldn't it be better to replace this with a dot plot given the small number of points?
6. The paper features many long paragraphs (e.g., pages 3-4). Would it be easier to read if the first paragraph was broken at "Contemporary recommendations" and "The potential role"? Same for the next paragraph which spans more than one page. Similarly see the first paragraph of section 2.2 which is very difficult to parse (possible add subsections for points 1-6?)
7. The "Effective treatment regimens" sentence is hard to parse: it mixes and matches therapies and disorders.

Review response letter

Reviewer #1

Q 1.1: “Using real-world Swedish registry data across the 21 Swedish regions (2016-2020) and implementing rigorous procedures to reduce putative confounding – in a sample encompassing 632 confirmed suicide deaths (200 of which in 15-19-year-olds) –, in the present study the Author investigated associations between regional adolescent excess suicide mortality (AESM) and treatment usage frequencies of clozapine, ECT and lithium (CEL) in 15-19-year-olds.

Overall, I found this study timely, original, well conducted and scientifically sound. The Authors incorporated all my suggestions in the revised paper. I have no further comments or concerns on it. I recommend acceptance.”

R 1.1: We thank you for these kind comments and are grateful for the astute suggestions provided throughout the review process!

Reviewer #2

Q 2.1: “I appreciate the authors' responses to reviewers' comments. Overall, I think the authors sufficiently addressed my concerns. This manuscript presents important information with clinical and potential policy relevance.”

R 2.1: We thank you for these comments and your work throughout the review process. We believe that your comments contributed to substantial improvements to the final version of the manuscript!

Q 2.2: “The authors provide a detailed reply to my comment, "I also encourage the authors include more cautions regarding the correlative nature of their study. The suicide prevention potential of these therapeutics cannot be arbitrated by this correlative study." In the authors' reply, they note that "we are concerned that the reviewer's comments may be overstated." I respectfully disagree. My comment was not that observational data do not have value. Instead, my comment was meant to highlight that the authors -- in my opinion -- needed to do a better job of contextualizing for the reader the methodology of their study and the claims made thereon. This is especially important for a prominent outlet that garners substantial press coverage. I maintain that a single correlative study cannot arbitrate the suicide prevention potential of these therapeutics. No single study is dispositive, regardless of methodology. I offer this additional context so that the authors and editors know that I do think their findings

have value. In their revision, I believe the authors adequately contextualized their findings. Thank you to the authors for this undertaking.”

R 2.2: We wholeheartedly agree that it is important to present the findings and interpretations in the context of the methodology utilized. Further, we acknowledge that the findings from our single study do not provide final and irrefutable evidence regarding the suicide preventive potential of the treatment modalities studied. Our previous response to the point in question could thus in part have been influenced by a misinterpretation on our part of the reviewer comment in question. We do however feel that we were alerted to the need to be more careful in the manner in which we presented and discussed our results. We thus want to express our gratitude for this and our happiness in learning that you found the changes made satisfactory.

Reviewer #3

Q 3.1.: “The authors have undertaken an analysis of $n=21$ regions in Sweden to identify associations between a collection of treatments (clozapine, ECT, and lithium usage rates) and suicide mortality in those regions. The authors have clearly noted the limitations of the study since they are unable to identify what overlap there is between the treatments and the outcomes because both rates arise from aggregated data. A creative approach to partially account for potential confounding is undertaken by subtracting the rates of suicides in older adolescents (15-19 year olds) from those in young adulthood (20-24 year olds) [page 7, lines 165-169].

I have had the opportunity to review the manuscript and supplementary materials, including the code and data. Overall these materials, methods, data, and code were clear and comprehensive. One comment: the way that the data were made accessible in R involved cutting and pasting values which wasn't an ideal workflow.

I believe that the paper is a valuable contribution to the literature despite the serious limitations due to its design. My main comments relate to ways to improve the presentation and reporting.”

R 3.1: We thank you for this assessment and kind comments regarding our manuscript! Regarding the inconvenience associated with accessing the data in R, we should add that we utilized the unaltered dataset provided by the Swedish National Board of Health and Welfare. We realize that this format is somewhat cumbersome, but, for transparency, we wanted to provide the dataset in a completely untransformed form.

Q 3.2.: “1. Title: I would encourage to authors to consider adding “Regional” before “Clozapine, ECT, and Lithium Usage Rates” in the title.”

R 3.2.: We have altered the title accordingly. Thank you for this suggestion.

Q 3.3.: “2. Reporting of p-values: I would encourage the authors to report p-values with two or three digits unless they are very small (e.g., $p < 0.0001$) rather than use $p < 0.5$ or $p > 0.10$ or $p = 0.04742$. Same for the Beta on page 11 (Beta = -0.4040600).”

R 3.3.: We have implemented this reporting style for p-values in the updated manuscript. Thank you!

Q 3.4.: “3. Reporting of sample size: it’s important for the authors to clearly remind the reader that this analysis is effectively $n = 21$ (despite the large populations involved).”

R. 3.4.: We have updated the methods section of the manuscript to convey this more clearly to the reader. Thanks for pointing this out!

Q 3.5.: “4. Discussion: add “regional” between “excess” and “suicide death rates”, page 13.”

R 3.5: Thank you. We have made this alteration to the revised manuscript.

Q 3.6: “5. Figure 2 would be improved if the axis labels included the sample size (e.g, Q1: Both sexes, $n = XX$). The violin plots are misleading here: wouldn’t it be better to replace this with a dot plot given the small number of points?”

R 3.6. Thanks for the suggested improvements to figure 2! The updated version of figure 2 has been altered in the proposed manner.

Q 3.7: “6. The paper features many long paragraphs (e.g., pages 3-4). Would it be easier to read if the first paragraph was broken at “Contemporary recommendations” and “The potential role”? Same for the next paragraph which spans more than one page. Similarly see the first paragraph of section 2.2 which is very difficult to parse (possible add subsections for points 1-6?)”

R 3.7: Thank you for alerting us to this. In the revised version of the manuscript, we have broken up these paragraphs in a manner that hopefully makes the manuscript easier to follow.

Q 3.8: “7. The “Effective treatment regimens” sentence is hard to parse: it mixes and matches therapies and disorders.”

R 3.8: In the revised manuscript, we have altered this sentence to make it a clearer read.
Thank you!